# Coronary Artery Bypass Grafting Versus Percutaneous Coronary Intervention for Left Main Coronary Artery Disease—Long-Term Outcomes

**DOI:** 10.3390/jcm14165747

**Published:** 2025-08-14

**Authors:** Szymon Jonik, Karolina Gumiężna, Piotr Baruś, Radosław Wilimski, Mariusz Kuśmierczyk, Grzegorz Opolski, Marcin Grabowski, Janusz Kochman, Zenon Huczek, Tomasz Mazurek

**Affiliations:** 11st Department of Cardiology, Medical University of Warsaw, Banacha 1a Str., 01-267 Warsaw, Poland; karolina.gumiezna@wum.edu.pl (K.G.); piotr.barus@wum.edu.pl (P.B.); grzegorz.opolski@wum.edu.pl (G.O.); marcin.grabowski@wum.edu.pl (M.G.); zenon.huczek@wum.edu.pl (Z.H.); tomasz.mazurek@wum.edu.pl (T.M.); 2Department of Heart, Thorax Surgery and Transplantology, Medical University of Warsaw, Banacha 1a Str., 01-267 Warsaw, Poland; rwilimski@gmail.com (R.W.); mariusz.kusmierczyk@wum.edu.pl (M.K.)

**Keywords:** multivessel coronary artery disease, coronary artery disease, percutaneous coronary intervention, Heart Team, left main coronary artery disease

## Abstract

**Background**: The optimal revascularization strategy for patients with left main coronary artery (LMCA) disease has been repeatedly addressed in randomized controlled trials (RCTs), although outcomes from real-life clinical studies are still poorly investigated. **Objectives**: This retrospective study aimed to assess the complete 5-year outcomes for individuals with multivessel coronary artery disease (MVD) involving LMCA disease treated with coronary artery bypass grafting (CABG) or percutaneous coronary intervention (PCI) as recommended by a local HT. **Methods**: From 2016 to 2019, 176 Heart Team (HT) meetings were held. Primary and secondary endpoints of 267 patients with MVD involving LMCA disease qualified either for CABG or PCI (109 and 158 patients, respectively) with subsequent optimal medical therapy (OMT) were assessed. The primary endpoint of the study was as an overall mortality, while secondary endpoints contained major adverse cardiac and cerebrovascular events (MACCE)—specifically, stroke, myocardial infarction (MI), repeat revascularization (RR), and the individual components of MACCE. **Results**: At 5 years, we found no significant difference in overall mortality between the both cohorts (22.9%-CABG vs. 24.7%-PCI, *p* = 0.74). The rate of MI was higher in patients treated percutaneously (7.3% vs. 15.8% for PCI, *p* = 0.04), while the incidence of stroke was higher in patients who underwent CABG (3.8% vs. 11.0% for CABG, *p* = 0.02). A MACCE occurrence was higher in PCI cohort (77.2% vs. 55.0%, *p* < 0.001), mainly driven by higher rates of RR was higher in patients treated percutaneously (32.9% vs. 13.8%, *p* < 0.001). **Conclusions:** For patients with LMCA disease, neither CABG nor PCI following HT decisions showed overwhelming superiority in real-life clinical practice: occurrence of all-cause death was similar, rates of MACCE, MI, and repeat revascularization advocated CABG, while incidence of strokes favored PCI.

## 1. Introduction

Left main coronary artery (LMCA) disease represents the highest-risk lesion of coronary artery disease (CAD), affecting a large myocardial area. Patients with significant left main (LM) stenosis are particularly vulnerable to increased risk of overall mortality, cardiovascular (CV) death and other major adverse cardiac and cerebrovascular events (MACCE) [1]. LMCA disease often occurs with concomitant multivessel disease (MVD), complicating treatment [2]. Guidelines consistently recommend surgical or percutaneous revascularization over optimal medical therapy (OMT) for this population, independently of clinical manifestation—even in asymptomatic or mildly symptomatic patients. For many years, coronary artery bypass grafting (CABG) has been a treatment of choice due to its established efficacy in providing complete and durable effect [3]. However, enormous advancement in percutaneous techniques has made it a technically viable, less invasive alternative [4,5]. Outcomes from clinical practice have been confirmed in several randomized controlled trials (RCTs), which demonstrated that patients with LMCA disease could be effectively treated with PCI with stent placement achieving similar long-term survival compared to CABG [6,7,8,9]. Despite similar mortality rates, both techniques differ significantly in the incidence of some MAACE components, which has led to ongoing debate regarding superiority of a given approach [6,10]. There exist several observational studies that address this issue as well. However, they were performed on specific populations including one large analysis of a Japanese cohort and the other enrolling the Canadian population [11,12]. Moreover, although they were performed relatively recently, the dynamics of development in invasive cardiology are such that the validity of studies for which patients began to be enrolled (as in the papers presented) in 2007 or 2008 may be less relevant now. In the presented manuscript, in contrast to other researchers, we also underline the importance of HT, and only included patients assessed by the multidisciplinary group of specialists. The current European Society of Cardiology (ESC) guidelines recommend using the SYNTAX II scale to guide revascularization decisions in LMCA disease, reflecting its prognostic value for long-term MACCE and overall death within the PCI-treated cohort, especially in individuals with low lesion complexity [3]. Yet, the best approach for patients with intermediate or high SYNTAX II scores is still debated. Another important issue is the fact that many of the patients we treat on a daily basis (i.e., the elderly, frail, or those with active cancer) are often excluded from ongoing clinical trials due to limitations in inclusion and exclusion criteria. This highlights the need for real-life studies to inform treatment strategies, particularly for patients typically excluded from RCTs due to stringent criteria. This approach represented by Heart Team (HT) is a key in the extensive management of the most complex cases of CAD, ensuring that even the most intricate scenarios receive tailored care, resulting in improved outcomes. The aim of our study was to parallel the 60-month outcomes between percutaneous and surgery revascularization in real-life individuals with MVD involving LMCA stenosis.

## 2. Materials and Methods

This single-center retrospective investigation was carried out at the 1st Department of Cardiology, Medical University of Warsaw, a large tertiary cardiovascular care center in Poland. We included patients aged ≥ 18 years with complete clinical, echocardiographic, and angiographic data. Patients with pregnancy/lactation, disseminated neoplastic process, life expectancy < 1 year, and lack of informed, written consent were excluded. Frailty was defined according to Cardiovascular Health Study criteria and assessed using Rockwood Frailty Scale (each patient was assessed personally). MVD was defined as occlusion angiographically greater than 70% in diameter (or between 40 and 70% but haemodynamically significant) in two—2-vessel disease (2-VD) or three—3-vessel disease (3-VD) –major coronary arteries or/and stenosis equal or more than 50% in diameter in LM—LMCA disease. All patients were evaluated weekly by HT composed of clinical cardiologists, cardiac surgeons, interventional cardiologists, and echocardiography specialists, and assigned for CABG or PCI with subsequent OMT. The decision by HT was made based on ESC Guidelines and other evidence-based data, also taking into account patients’ preferences, their clinical, laboratory, and echocardiographic features, technical feasibility, and risk assessment. All patients presented during the Heart Team meetings were provided with a personalized patient history file containing their medical history, coronary angiography, echocardiography, and laboratory test results. Each Heart Team member had access to this file and the opportunity to review it. Subsequently, after the presentation of the angiography and echocardiography results, the Heart Team members discussed and made a decision regarding eligibility. Only a unanimous decision by all Heart Team members constituted final approval for further treatment. Heart Team meetings were prospective, while the analysis was retrospective. Out of 1509 patients consulted during 176 HT councils in years 2016–2019, 1035 were qualified for invasive strategy. A total number of 267 individuals with MVD involving LMCA disease were selected for this sub analysis, including 109 patients qualified for CABG and 158 for PCI. OMT was stated as the implementation of medications with a proven effect on prognosis or angina symptoms. We determined primary endpoints as overall mortality, while secondary endpoints contained major adverse cardiac and cerebrovascular events (MACCE)—specifically, stroke, myocardial infarction (MI), repeat revascularization (RR), and the individual components of MACCE. The median (Q1; Q3, SD) follow-up was 52.3 [36.4; 66.1] months. The terms of chronic kidney disease (CKD), severe pulmonary arterial hypertension (PAH), and anemia have been described earlier [13]. The RR refers to unplanned procedures and also planned staged PCIs—both target-vessel or target-lesion driven, both clinically and angiographic driven. The main outline of the study was presented in Figure 1. 

### Statistical Analysis

The statistical analysis was performed using PQStat software (version 1.6.6, PQStat, Poznań, Poland). The normality of distribution for continuous variables was assessed using the Shapiro–Wilk test. Categorical data were expressed as counts and percentages, while continuous data as means with standard deviation (SD) or as median with 1st and 3rd quartile (Q1, Q3), adequately to their distribution. Comparisons between groups were performed using the Pearson’s Chi-squared test for categorical variables, and Student’s *t* test or U Mann–Whitney test for continuous variables, according to data distribution. To compare the risk of primary endpoint between study groups, the relative risk (RR) with 95% confidence intervals (95% CI) were estimated. We used the Kaplan–Meier method with a log-rank test and Cox proportional hazards model to determined time to event analysis. Furthermore, given the differences in the baseline clinical and angiographic characteristics between both cohorts, a propensity score matching analysis was used to equalize the baseline variables (or characteristics) between the groups with PCI or CABG. The propensity score was the conditional probability of experiencing a specific exposure (percutaneous versus surgery revascularization) given a set of covariates measured at baseline. The propensity score was estimated using a non-parsimonious multivariate logistic regression model, with PCI as the dependent variable and all items from baseline described in Appendix A as covariates. The selection of participants for the control group in propensity score matching analysis was conducted using the balanced nearest neighbor method (therefore, matching the most similar individuals, i.e., with the closest propensity score value). In the matched cohort, paired comparisons were performed with the use of McNemar’s test for binary variables and a paired Student’s *t*-test for continuous variables. Both overall mortality and MACCE were analyzed using the Cox model as separate dependent variables. Variance Inflation Factor (VIF) was used to detect multicollinearity among predictor variables. Variables with VIF values less than 5 are considered acceptable. Among parameters from clinical, echocardiographic or angiographic variables, 10 showed significant relationship with overall mortality: age, frailty, COPD, diabetes treated with insulin, anemia, PH, PAD, CKD, complete revascularization, and total occlusion. Among parameters from clinical, echocardiographic or angiographic variables, 9 showed significant relationship with rates of MACCE: frailty, COPD, congestive heart failure, diabetes, PH, previous stroke/TIA, prior revascularization, PAD and complete revascularization. A two-sided *p* value of less than 0.05 indicated statistical significance.

## 3. Results Study Population

In the years 2016 and 2019, 176 HT meetings were conducted, resulting in identifying 267 individuals with LMCA occlusion fulfilling inclusion and exclusion criteria. All individuals were qualified by HT for surgery or percutaneous stenting with following OMT and were observed for a median [Q1; Q3] of 52.3 [36.4; 66.1] months. The median (Q1; Q3) age was 74 (67; 78) years, 75.3% male, and 17.2% presented with frailty syndrome. The median (Q1; Q3) preprocedural risk scores were 5.8 (3.6; 9.9) for EuroSCORE II [European System for Cardiac Operative Risk Evaluation II] and 3.7 (2.3; 6.4) for STS [Society of Thoracic Surgeons]. Approximately 42% had medically treated diabetes, nearly half of whom received insulin. The prevalence of CAD risk factors including hypertension, dyslipidemia (and current smoking was high. Notably, significant comorbidity differences between cohorts included chronic kidney disease CKD, anemia, dyslipidemia, AF and active cancer, being higher in PCI group. Complete revascularization was performed in 55.8% of patients, more frequently in surgery groups. The variables from baseline clinical characteristics are presented in Table 1.

### 3.1. Angiographic Parameters

The CAD burden was relatively high in the study population. Most patients (97%) had LMCA segment disease, while 3% of them had equivalent of LMCA disease, defined as stenosis of 50% or more in both the ostial LAD (left anterior descending) and ostial LCx (left circumflex) coronary arteries, assessed by quantitative coronary angiography. Distal bifurcation or trifurcation involvement of LM was seen in 85% of cases. The prevalence of affected non-LM arteries was well-balanced between cohorts—with 2-VD being the most frequent concomitant. A total of 32.2% of lesions were severely calcified (densities visible without heart motion and typically affecting both sides of the vessel), and in 26.2% of individuals at least one artery was chronically occluded. LM as culprit lesion was considered in 17/108 (15.7%) of patients qualified for PCI and 12/66 (18.2%) of individuals qualified for CABG; *p* = 0.68. The variables from angiographic parameters were detailed in Table 2.

### 3.2. Medications on Admission and at Discharge

Generally, adherence to OMT was satisfactory, although drugs usage differed between the groups during follow-up—Table 3.

### 3.3. Outcomes

Five-year follow-up was achieved in 100% of patients. Overall mortality rates did not significantly differ between the CABG and PCI groups (25/109 (22.9%) vs. 39/158 (24.7%), *p* = 0.74)—Figure 2. The PCI group had higher rates of MACCE, mainly driven by higher rates of repeat revascularization (122/158 (77.2%) vs. 60/109 (55.0%) for CABG, *p* < 0.001 and 52/158 (32.9%) vs. 15/109 (13.8%) for CABG, *p* < 0.001, respectively). The incidence of MI was significantly higher in PCI cohort (25/158 (15.8%) vs. 8/109 (7.3%) for CABG, *p* = 0.04), while stroke occurred more often in patients treated surgically (12/109 (11.0%) vs. 6/158 (3.8%) for PCI, *p* = 0.02). In-hospital mortality was comparable between CABG and PCI cohorts (5/109 (4.6%) vs. 6/158 (3.8%), respectively, *p* = 0.75). Post-interventional in-hospital stay was significantly prolonged in surgery cohort—median (Q1; Q3): 6 (4;11) vs. 1 (1;3) days for PCI, respectively; *p* < 0.001. The outcomes comparing surgery and percutaneous stenting were detailed in Table 4. The outcomes for the primary endpoint in prespecified subgroups were shown in Figure 3.

### 3.4. Propensity Score Matching Analysis

We used propensity score matching (PSM) to assess individuals with similar baseline characteristics. From PCI group 53 patients were selected based on matching and compared to 53 patients from CABG cohort as reference—Appendix A. Primary and secondary outcomes in the propensity score matched cohorts were presented in Table 5, revealing that in 5-year follow-up PCI was associated with higher incidence of repeat revascularization, while CABG-treated patients had significantly prolonged postprocedural hospital stay. No differences in rates of all-cause death, MACCE, MI, strokes, and in-hospital stay were observed in matched cohorts.

### 3.5. Multivariable Cox Proportional Hazards Model

Among parameters from clinical, echocardiographic or angiographic variables, 10 showed a significant relationship with overall mortality: (1) age, per 1 year increase [HR (95% CI)]: 1.10 (1.05–1.12), *p* = 0.003; (2) frailty [HR (95% CI)]: 3.92 (2.55–7.34), *p* < 0.001; (3) COPD [HR (95% CI)]: 4.56 (2.92–8.11), *p* < 0.001; (4) diabetes treated with insulin [HR (95% CI)]: 2.79 (1.63–4.95), *p* < 0.001; (5) anemia [HR (95% CI)]: 1.45 (1.11–2.03), *p* = 0.002; (6) PH [HR (95% CI)]: 5.01 (3.02–9.14), *p* < 0.001; (7) PAD [HR (95% CI)]: 2.34 (1.48–4.09), *p* < 0.001; (8) CKD [HR (95% CI)]: 2.09 (1.35–3.77), *p* < 0.001; (9) complete revascularization [HR (95% CI)]: 2.99 (1.70–5.12), *p* < 0.001; and (10) total occlusion [HR (95% CI)]: 1.85 (1.28–2.70), *p* < 0.001. Among parameters from clinical, echocardiographic or angiographic variables, 9 showed significant relationship with rates of MACCE: (1) frailty [HR (95% CI)]: 2.65 (1.71–4.56), *p* < 0.001; (2) COPD [HR (95% CI)]: 3.12 (2.05–6.96); (3) congestive heart failure [HR (95% CI)]: 1.08 (1.02–1.15), *p* < 0.04; (4) diabetes [HR (95% CI)]: 2.48 (1.66–3.97), *p* < 0.001; (5) PH [HR (95% CI)]: 4.01 (2.46–7.75), *p* < 0.001; (6) previous stroke/TIA [HR (95% CI)]: 1.89 (1.41–3.38), *p* < 0.001; (7) prior revascularization [HR (95% CI)]: 2.80 (1.72–4.15), *p* < 0.001; (8) PAD [HR (95% CI)]: 1.95 (1.26–3.64), *p* < 0.001; and (9) complete revascularization [HR (95% CI)]: 2.28 (1.43–4.10), *p* < 0.001.

## 4. Discussion

### 4.1. Main Findings

In our real-life, retrospective study we focused on long-term outcomes of patients with MVD involving LMCA disease qualified by HT to CABG or PCI followed by long-term OMT. No CABG nor PCI demonstrated significant superiority in terms of all-cause mortality. The incidence of MI, repeat revascularization, and MACCE were higher in patients who underwent PCI, whereas rates of stroke and prolonged postprocedural hospital stay penalized CABG.

### 4.2. Interpretation of Main Findings

As this is an observational study, reflective of ordinary clinical practice, differences in clinical variables observed between PCI and CABG cohorts were expected. However, in the analysis of the outcomes for the primary endpoint in prespecified subgroups, none of the clinical or angiographic parameters indicated superiority of given approach. For most of them surgery showed predominance, although not statistically significant. Despite these observations, some differences in characteristics or outcomes warrant further consideration. Given that patients undergoing percutaneous stenting were typically older, more frail and more burdened, comparable mortality between both cohorts might be considered as hypothesis–generating. This may be attributed to significant advancement in PCI techniques, including new generation of stents, better quality and accessibility of intravascular imaging, improved procedural expertise, and the individualization of antiplatelet therapy.

### 4.3. Other Studies in the Field and Novelty of the Study

Despite the above, the incidence of MACCE was higher in the PCI group, driven by more frequent occurrences of MI and repeat revascularization in the long-term follow-up, aligning partially with our initial assumptions and previous studies [14]. This phenomenon could be explained by the higher rate of complete revascularization achieved in the surgery cohort (70.6%) compared to those who underwent percutaneous stenting (45.6%). Also, the higher rates of complete revascularization in CABG cohort were likely associated with higher MI- and RR- rates in long-term follow-up in the PCI group. This difference is higher than in previous studies, owing mainly to a relatively low rate of complete revascularization reached for PCI-treated patients in our analysis. Percutaneous stenting was superior to surgery regarding the postprocedural hospital stay and the rates of strokes, nearly four- and three- times lower than in CABG cohort, respectively. The shorter hospitalization duration in PCI patients was expected due to being much less invasive, and it does not require general anesthesia. The risks of perioperative infection have also decreased. However, in-hospital mortality did not differ between both cohorts.

Interestingly, we found that HT’s decision on the optimal treatment for CAD has increasingly depended more on the clinical features of the patient rather than solely on angiographic findings. This shift might be due to significant technical advancements in percutaneous techniques that have expanded the capabilities of PCI to address more complex coronary artery anatomies, traditionally deemed only suitable for CABG. Innovations such as drug-eluting stents, advanced imaging modalities, and improved procedural techniques have enhanced the success rates and long-term outcomes of PCI, making it a viable option for a broader range of patients.

In several RCTs involving patients with LMCA disease, percutaneous stenting showed more favorable outcomes at 1 year compared to surgery, including fewer periprocedural adverse events and faster recovery [13,15,16]. However, these trials presented varying long-term outcomes [13,16,17]. Regarding short-term follow-up, in the NOBLE trial, the main differences between study groups in 30-day outcomes included more frequent bleedings, need for blood transfusion, strokes, and longer hospital stay in the CABG group [16]. However, the difference in early all-cause death and CV mortality was not statistically significant between studied cohorts, aligning with our results. Similarly, the EXCEL trial found a non-significantly higher overall mortality in CABG cohort, though the risk of stroke and MI in the first 30 days after the procedure was increased [6]. An elevated incidence of strokes in the CABG group, which may seem intuitive given the less invasive nature of the PCI procedure, is consistent with some previous research indicating an elevated risk of stroke associated with surgery [14,18].

All above findings emphasize the importance of balancing the risks and benefits of each revascularization strategy and highlight the need for careful perioperative management. Evaluating the anatomical complexity of CAD, commonly assessed using the SYNTAX II score, is crucial. In our real-life sub analysis, most included individuals had at least an intermediate SYNTAX II score, reaching the mean of 30.4 in overall population. Generally, the higher the SYNTAX score, the greater the likelihood that the patient will be qualified for CABG, although the qualification score itself depends on many factors related to chronic comorbidities and other angiographic parameters, not just the SYNTAX score.

Available evidence indicates that patients with LMCA disease and low anatomic complexity of coronary lesions achieve similar outcomes regardless of the choice of revascularization strategy chosen [19], which was reflected in the current ESC guidelines [3]. Generally, surgery is preferred for patients with intermediate or high SYNTAX II scores, based on analyses from trials involving SYNTAX score [20,21], which demonstrated better outcomes of surgery at higher complexity levels [3]. However, the PRECOMBAT trial did not confirm these results, showing no statistically significant differences in endpoints according to treatment across SYNTAX score tertiles in both 5- and 10-years of follow-up [8,22]. It was possibly related to the limited statistical power, but these results forced us to vigilantly verify the subject. Notably, the same study, including patients with both LMCA disease and 3-VD, showed that CABG was superior to PCI regarding the rates of MACCE [22]. On the other hand, the 5-year follow-up in NOBLE trial suggested the superiority of surgery regarding MACCE, irrespective of anatomical complexity. However, in this study very complex cases with more than three additional non-complex lesions were excluded [16]. A recent meta-analysis by Sabatine MS et al. covering most important RCTs to date showed no significant differences in mortality between groups treated with surgery or PCI, while key secondary outcomes, including spontaneous MI and repeat revascularization, were significantly increased with PCI [14]. However, we must keep in mind that most individuals included had low or intermediate lesion complexity (median SYNTAX score of 34.0 (28.0; 38.8)), making it difficult to apply these results to patients with the most advanced CAD. Furthermore, the coexistence of MVD added complexity to CAD management, favoring CABG over PCI regarding all-cause mortality both in 5- and 10-year follow-ups [9,23], which was confirmed in a pooled analysis summarizing the largest RCTs to date [24].

Also, several observational studies examined this issue. PCI and CABG in patients with LMD, among others, were compared in two large Asian registries—Japan and Korea. In both studies, percutaneous stenting showed similar rates of death and primary composite outcomes, but an elevated incidence of coronary revascularization at long-term follow-up [11,25]. To our knowledge, there are no comparable studies in the European population. In addition, a great advantage of the presented work over previous ones is the recruitment of patients in 2016–2019, which makes it much more relevant. Moreover, the study of a Polish population showed that the risk of PCI is more related to the patient’s baseline burden than to the lesion’s characteristics or total stent length alone [26]. This is valuable information in the context of MVD patients requiring multiple stents. In addition, it sets high the position of HT, which considers all the aggravating factors and not just the characteristics of the diseased vessel lesion.

Another issue that should be discussed is the management of patients with ACS and the mode of HT-action in these individuals. It is logical that in this cohort the further proceeding hugely depends on the patient’s condition, and the decision is often individual. It is worth noting, in our study, that a high percentage of participants presented with ACS-65.2% (CABG-60.6%, PCI-68.4%), and hence, the results of the final analysis represent a mixture of stable and unstable patients. Obviously, patients with STEMI or high-risk NSTEMI were treated immediately after coronary angiography if the culprit lesion was found.

### 4.4. Interpretation of the Main Findings

There are several components making comparisons between surgery and percutaneous stenting challenging: (1) significant differences in baseline clinical and angiographic characteristics; (2) unequal number of patients in studied cohorts; and (3) the influence of specialists’ experience and available methods on the selection of the revascularization strategy. Despite these challenges, it is crucial to examine revascularization mode in relation to outcomes in observational studies. While the results from RCTs are widely available, the exclusion of many patients managed within daily clinical practice diminishes their importance, and thus non-randomized research should be considered as complementary to RCTs. A great advantage of our investigation is the enrollment of real-life all-comer patients and their assignment to treatment strategy after careful HT evaluation, while in previous trials, a significant percentage of patients with LMCA occlusion were disqualified due to lack of consent or unfulfillment of inclusion/exclusion criteria [27,28]. This approach may more reliably translate into different outcomes. The analysis based on a real-world population with LMCA disease from DELTA-2 registry showed that nearly 40% of patients would be excluded from the EXCEL trial [19], demonstrating statistically different results for potentially included and excluded groups [29].

Nowadays, multidisciplinary HT management and the individualization of a percutaneous approach based on novel intracoronary imaging tools should be included into the comprehensive evaluation of patients with LMCA disease. The involvement of HT in the decision-making process for the severely burdened individuals should be considered to determine the most appropriate treatment strategy [30]. Furthermore, the modern and effective percutaneous stenting of patients with LMCA disease cannot occur without widespread use of intra-vascular ultrasound (IVUS), optical coherence tomography (OCT) or fractional flow reserve (FFR). The high utilization of intracoronary imaging modalities in our study may have translated into a high rate of successful percutaneous interventions. PCI outcomes in other centers with lower rates of intracoronary imaging usage would likely have been worse. The lack or rare usage of modern intracoronary imaging tools could also translate into higher rates of repeat revascularization in PCI patients. Furthermore, sub analyses of the EXCEL and SYNTAX trials suggest that unplanned repeat revascularization may be associated with worsened survival and quality of life [27,31]. In our study, according to current guidelines, we also performed the vast majority of PCI of LM with the utilization of intracoronary imaging—see Results and Table 2.

Additionally, in a recently published article regarding one-year prognostic differences and management strategies between ST-elevation and non-ST-elevation myocardial infarction at 1-year follow-up, overall outcomes were comparable between groups. Nonfatal reinfarction occurred more frequently in patients with NSTEMI (3.4% versus 2.8%, *p* = 0.022), but this association was not significant after adjustment (odds ratio [OR] 0.90, 95% confidence interval [CI] 0.65–1.24, *p* = 0.519). Results from propensity score matched analyses confirmed the absence of prognostic differences. Subgroup analyses revealed significant interactions for diabetes mellitus and completeness of revascularization. As the authors of the article emphasized that 1-year outcomes were largely similar in patients with STEMI and NSTEMI. Differences in reinfarction risk appear to be driven by baseline characteristics and treatment patterns, rather than infarct type itself [32].

### 4.5. Limitations and Conclusions

To summarize, in view of previous studies, the aim of the overarching goal still exists. We strongly believe that the conclusion regarding optimal treatment strategy should often be made on a case-by-case basis, considering the decision of the HT, as well as the patient’s preferences. Further studies should focus on subgroups highly burdened with comorbidities and the different outcomes upon implemented revascularization strategies.

## 5. Limitations

Despite advantages of this study, it should be considered with some limitations. The main one is its nonrandomized character. Even though we included appropriate patients with accuracy, the risk of bias could be higher than within RCTs. Moreover, the group size was relatively small when compared to previous trials. We should highlight the significant differences between baseline characteristics of both study groups, although these disproportions were expected related to the nature of this research. The HT evaluations may be one-specific and do not reflect decisions made in other hospitals. The experience of surgical and percutaneous teams cannot be overlooked. The regular use of medications cannot be objectively measured. Moreover, a hierarchy within HT also exists, which can lead to bias as well [33].

## 6. Conclusions

In our single-center observational study we demonstrated that although an overall mortality did not significantly differ between treatment cohorts, some disparities in secondary endpoints, as revealed in previous RCTs, were observed. An optimal choice of revascularization strategy should never be individual and the implementation of HT into decision process is highly required. Further studies including RCTs assessing the role of HT are essential yet the role of real-life population analyses should be underscored.

## Figures and Tables

**Figure 1 jcm-14-05747-f001:**
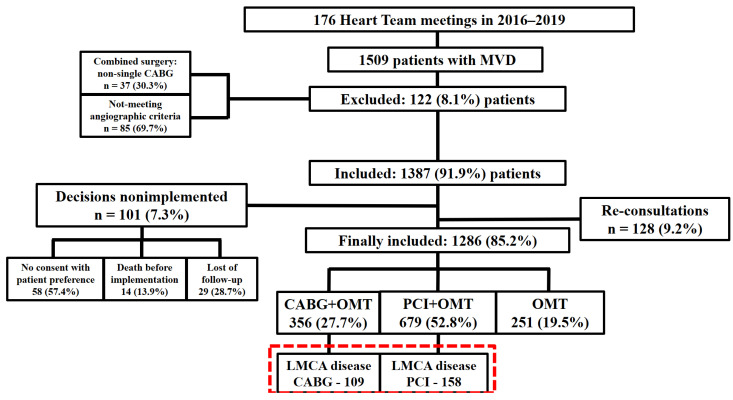
**The main outline of the study. CABG**—coronary artery bypass grafting; **LMCA**—left main coronary artery; **MVD**—multivessel coronary artery disease; **OMT**—optimal medical therapy; **PCI**—percutaneous coronary intervention.

**Figure 2 jcm-14-05747-f002:**
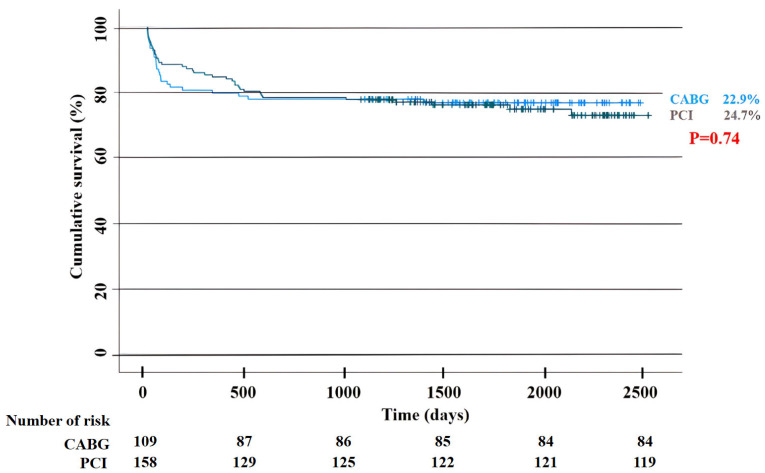
**The overall mortality in both cohorts. CABG**—coronary artery bypass grafting; **PCI**—percutaneous coronary intervention.

**Figure 3 jcm-14-05747-f003:**
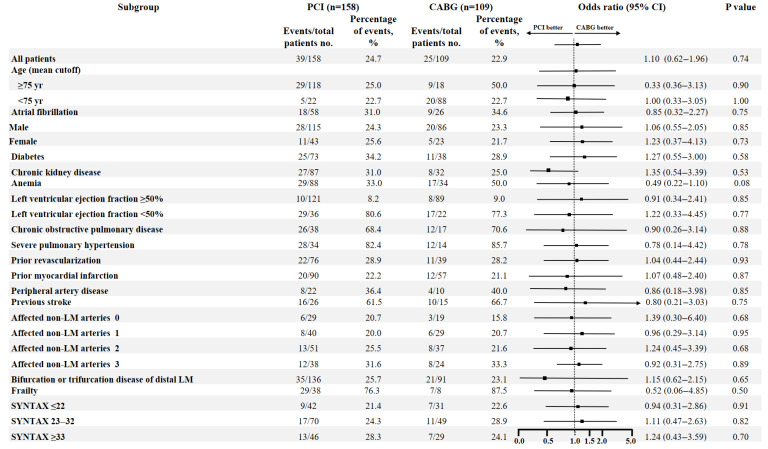
**Subgroup analyses of the primary endpoint at 5 years. CABG**—coronary artery bypass grafting; **LM**—left main; **PCI**—percutaneous coronary intervention.

**Table 1 jcm-14-05747-t001:** **Baseline clinical characteristics. ACS**—acute coronary syndrome; **BMI**—body mass index; **CABG**—coronary artery bypass grafting; **CCS**—Canadian Cardiovascular Society; **CKD**—chronic kidney disease; **COPD**—chronic obstructive pulmonary disease; **EuroSCORE II**—European System for Cardiac Operative Risk Evaluation II; **LVEF**—left ventricle ejection fraction; **MI**—myocardial infarction; **PAD**—peripheral artery disease; **PCI**—percutaneous coronary intervention; **PH**—pulmonary hypertension; **STEMI**—ST-segment elevation myocardial infarction; **STS**—Society of Thoracic Surgeons score; **TIA**—transient ischemic attack; **UA**—unstable angina.

	Characteristics	Overall (267)	CABG (109)	PCI (158)	*p* Value
**Preprocedural**	**Age, years; median** **(Q1; Q3)**	74 (67; 78)	71 (62; 76)	76 (69; 79)	<0.001
	**Gender, male; *n* (%)**	201 (75.3)	86 (78.9)	115 (72.8)	0.26
	**BMI, kg/m^2^; median (** **Q1; Q3)**	28.2 (26.3; 29.7)	28.2 (26.8; 29.0)	28.1 (25.6; 30.1)	0.92
	**Frailty; *n* (%)**	46 (17.2)	8 (7.3)	38 (24.1)	<0.001
	**Current smoking; *n* (%)**	68 (25.5)	24 (22.0)	44 (27.8)	0.28
	**COPD; *n* (%)**	55 (20.6)	17 (15.6)	38 (24.1)	0.09
	**Diabetes; *n* (%)**	111 (41.6)	38 (34.9)	73 (46.2)	0.07
	**with insulin; *n* (%)**	51 (19.1)	16 (14.7)	35 (22.2)	0.13
	**Hypertension; *n* (%)**	237 (88.8)	92 (84.4)	145 (91.8)	0.06
	**Severe PH; *n* (%)**	48 (18.0)	14 (12.8)	34 (21.5)	0.07
	**Dyslipidemia; *n* (%)**	214 (80.1)	80 (73.4)	134 (84.8)	0.02
	**Congestive heart failure; *n* (%)**	58 (21.7)	22 (20.2)	36 (22.8)	0.61
	**LVEF, %; median (Q1; Q3)**	34.0 (30.0; 50.0)	34.0 (31.0; 50.0)	33.0 (29.0; 50.0)	0.24
	**CKD; *n* (%)**	119 (44.6)	32 (29.4)	87 (55.1)	<0.001
	**Atrial fibrillation; *n* (%)**	84 (31.5)	26 (23.9)	58 (36.7)	0.03
	**Anemia; *n* (%)**	120 (44.9)	34 (31.2)	88 (55.7)	<0.001
	**Prior MI; *n* (%)**	147 (55.1)	57 (52.3)	90 (57.0)	0.45
	**Prior revascularization; *n* (%)**	115 (43.1)	39 (35.8)	76 (48.1)	0.05
	**Indication; *n* (%)**				
	**CCS**	93 (34.8)	43 (39.4)	50 (31.6)	0.19
	**ACS**	174 (65.2)	66 (60.6)	108 (68.4)	0.19
	**STEMI**	11 (4.1)	2 (1.8)	9 (5.7)	0.12
	**NSTEMI/UA**	163 (61.0)	64 (58.7)	99 (62.7)	0.52
	**PAD; *n* (%)**	32 (12.0)	10 (9.2)	22 (13.9)	0.24
	**Prior stroke/TIA; *n* (%)**	41 (15.4)	15 (13.8)	26 (16.5)	0.55
	**Active cancer; *n* (%)**	15 (5.6)	0 (0.0)	15 (9.5)	<0.001
	**EuroSCORE II, %; median (Q1; Q3)**	5.8 (3.6; 9.9)	3.8 (3.2; 5.3)	9.8 (6.0; 10.5)	<0.001
	**STS score, %; median (Q1; Q3)**	3.7 (2.3; 6.4)	2.5 (2.1; 3.3)	6.1 (3.9; 6.7)	<0.001
**Procedural**	**Complete revascularization; *n* (%)**	149 (55.8)	77 (70.6)	72 (45.6)	<0.001

**Table 2 jcm-14-05747-t002:** **Angiographic parameters. CABG**—coronary artery bypass grafting; **LM**—left main; **PCI**—percutaneous coronary intervention; **SYNTAX**—Synergy between PCI with Taxus and Cardiac Surgery; *** LM equivalent disease**—diameter stenosis of both the ostial **LAD** (left anterior descending) and ostial **LCx** (left circumflex) coronary arteries ≥50% by quantitative coronary angiography, without ≥50% LM diameter stenosis.

Angiographic Parameters	Overall (267)	CABG (109)	PCI (158)	*p* Value
**Qualifying LM lesion**				
**LM coronary segment**	259 (97.0)	106 (97.2)	153 (96.8)	0.85
**LM equivalent disease ***	8 (3.0)	3 (2.8)	5 (3.2)	0.85
**Bifurcation or trifurcation disease of distal LM; *n* (%)**	227 (85.0)	91 (83.5)	136 (86.1)	0.56
**LM considered as culprit lesion in ACS, *n* (%)**	29/174 (16.7)	12/66 (18.2)	17/108 (15.7)	0.68
**Affected non-LM arteries, *n* (%)**				
**0**	48 (18.0)	19 (17.4)	29 (18.4)	0.0971
**1**	69 (25.8)	29 (26.6)	40 (25.3)	
**2**	88 (33.0)	37 (33.9)	51 (32.3)	
**3**	62 (23.2)	24 (22.0)	38 (24.1)	
**Severe calcification; *n* (%)**	86 (32.2)	34 (31.2)	52 (32.9)	0.77
**Total occlusion; *n* (%)**	70 (26.2)	25 (22.9)	45 (28.5)	0.31
**Number of stents implanted per patient; median (Q1; Q3)**	NA	NA	2 (1;3)	NA
**Number of conduits (arterial and venous) per patient; median (Q1; Q3)**	NA	3 (2;3)	NA	NA
**IVUS utilization; *n* (%)**	NA	NA	154 (97.5)	NA
**FFR utilization; *n* (%)**	NA	NA	48 (30.4)	NA
**OCT utilization; *n* (%)**	NA	NA	23 (14.6)	NA
**Rotational atherectomy; *n* (%)**	NA	NA	11 (7.0)	NA
**SYNTAX score; median (Q1;Q3)**	34.0 (28.0; 38.8)	34.0 (29.0; 41.0)	33.8 (23.8; 37.5)	0.03

**Table 3 jcm-14-05747-t003:** **Medications on admission and at discharge. ACE**—angiotensin-converting enzyme; **ARB**—angiotensin receptor blocker; **CABG**—coronary artery bypass grafting; **P2Y12 inhibitors**—clopidogrel, prasugrel, ticagrelor; **PCI**—percutaneous coronary intervention.

**Medications on Admission**	**Overall (267)**	**CABG (109)**	**PCI (158)**	***p* Value**
**Statin; *n* (%)**	231 (86.5)	81 (74.3)	150 (94.9)	<0.001
**ACE inhibitor; *n* (%)**	196 (63.3)	62 (56.9)	107 (67.7)	0.07
**ARB; *n* (%)**	65 (24.3)	31 (28.4)	34 (21.5)	0.20
**Beta-blocker; *n* (%)**	208 (77.9)	85 (78.0)	123 (77.8)	0.98
**Medications at discharge**	**Overall (267)**	**CABG (109)**	**PCI (158)**	***p* value**
**Aspirin; *n* (%)**	234 (87.6)	101 (92.7)	133 (84.2)	0.04
**P2Y12 inhibitors; *n* (%)**	145 (54.3)	21 (19.3)	124 (78.5)	<0.001
**Clopidogrel, *n* (%)**	88 (60.7)	19 (90.5)	69 (55.6)	<0.001
**Ticagrelol, *n* (%)**	37 (25.5)	1 (4.8)	36 (29.0)	<0.001
**Prasugrel, *n* (%)**	20 (13.8)	1 (4.8)	19 (15.3)	<0.001
**Statin; *n* (%)**	243 (91.0)	91 (83.5)	152 (96.2)	<0.001
**ACE inhibitor; *n* (%)**	191 (71.5)	74 (67.9)	117 (74.1)	0.27
**ARB; *n* (%)**	70 (26.2)	29 (26.6)	41 (25.9)	0.91
**Beta-blocker; *n* (%)**	213 (79.8)	78 (71.6)	135 (85.4)	0.006
**Loop diuretic; *n* (%)**	144 (53.9)	80 (73.4)	64 (40.5)	<0.001
**Aldosterone antagonist; *n* (%)**	57 (21.3)	13 (11.9)	44 (27.8)	0.002

**Table 4 jcm-14-05747-t004:** **Primary and secondary endpoints. CI**—confidence interval; **CABG**—coronary artery bypass grafting; **MACCE**—major adverse cardiac and cerebrovascular event; **MI**—myocardial infarction; **PCI**—percutaneous coronary intervention; **RR**—relative risk.

Endpoints	Overall (267)	CABG (109)	PCI (158)	RR PCI vs. CABG (95% CI)	*p* Value
**All-cause death, *n* (%)**	64 (24.0)	25 (22.9)	39 (24.7)	1.08 (0.69–1.67)	0.74
**MACCE, *n* (%)**	182 (68.1)	60 (55.0)	122 (77.2)	1.40 (1.16–1.70)	**<0.001**
**MI, *n* (%)**	33 (12.4)	8 (7.3)	25 (15.8)	2.16 (1.01–4.60)	**0.04**
**Stroke, *n* (%)**	18 (6.7)	12 (11.0)	6 (3.8)	0.34 (0.13–0.89)	**0.02**
**Periprocedural, *n* (%)**	4 (1.5)	3 (2.8)	1 (0.6)	0.27 (0.10–0.75)	**0.01**
**Repeat revascularization, *n* (%)**	67 (25.1)	15 (13.8)	52 (32.9)	2.40 (1.45–4.10)	**<0.001**
**In-hospital mortality, *n* (%)**	11 (4.1)	5 (4.6)	6 (3.8)	0.83 (0.26–2.65)	0.75
**Postprocedural hospital stay, days; median (Q1; Q3)**	4 (1;7)	6 (4;11)	1 (1;3)	-	**<0.001**

**Table 5 jcm-14-05747-t005:** Primary and secondary outcomes in the propensity score matched cohorts. yr-year.

Outcome	Number of Patients with Event	Event Rate (%/yr)	Hazard Ratio (95% CI)	*p* Value
**All-cause death, *n* (%)**				
PCI	12/53 (22.6)	4.5	1.09 (0.43–2.75)	0.82
CABG	11/53 (20.8)	4.2	Reference	
**MACCE, *n* (%)**				
PCI	36/53 (67.9)	13.6	1.33 (0.60–2.94)	0.08
CABG	27/53 (50.9)	11.0	Reference	
**MI, *n* (%)**				
PCI	7/53 (13.2)	2.6	1.17 (0.48–6.37)	0.34
CABG	4/53 (7.5)	1.5	Reference	
**Stroke, *n* (%)**				
PCI	2/53 (3.8)	0.75	0.33 (0.06–1.73)	0.14
CABG	6/53 (11.3)	2.3	Reference	
**Repeat revascularization, *n* (%)**				
PCI	15/53 (28.3)	5.7	2.5 (0.88–7.06)	0.03
CABG	6/53 (11.3)	2.3	Reference	
**In-hospital mortality, *n* (%)**				
PCI	1/53 (1.9)	0.38	0.33 (0.03–3.31)	0.31
CABG	3/53 (5.7)	1.1	Reference	
**Postprocedural hospital stay, days; mean (SD)**				
PCI	1.6 (1.1)	-	-	<0.001
CABG	8.5 (4.4)	-	Reference	

## Data Availability

The data presented in this study are available on request from the corresponding author.

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
