# Peer review of "Coronary Artery Bypass Grafting Versus Percutaneous Coronary Intervention for Left Main Coronary Artery Disease—Long-Term Outcomes"

_jcm, 2025, doi:10.3390/jcm14165747_

Round 1

Reviewer 1 Report

Comments and Suggestions for Authors

Congratulations to the author for their manuscript; here my comments about it:

The authors should clearly describe how the SYNTAX scores influenced the treatment decision-making process in their methods

Although frailty was defined using the Cardiovascular Health Study criteria, the manuscript does not detail the scoring method or how these criteria were applied operationally. Authors should clarify this aspect

The definition of “repeat revascularization” requires clarification. It is not specified whether this includes only unplanned procedures or also planned staged PCI interventions.

Further details should be provided on how the Heart Team decisions were standardized, including whether a checklist, algorithm, or structured scoring system was employed.

The logistic regression model used for propensity score matching should list the specific covariates included, particularly whether the SYNTAX score was among them, which is not clear

The reported range for hospital stay appears inconsistent; for example, a median duration of 6 days with an interquartile range (IQR) of 8–11 days is statistically implausible, as the median should fall between the first and third quartiles. Authors should clarify this aspect

Moreover, authors are encouraged to include the latest evidence about the different prognosis of ACS patients (https://doi.org/10.1007/s40256-025-00739-8)

Author Response

Dear Reviewer

We would like to thank the Editors and the Reviewers for taking the time to evaluate this work and for providing valuable comments, which, as we believe, helped us to further improve the submitted manuscript.

The introduced changes were highlighted in the manuscript in yellow.

Your inquiry: The authors should clearly describe how the SYNTAX scores influenced the treatment decision-making process in their methods.

Our response: Overall, the SYNTAX score helped specialists participating in the Heart Team meetings to decide on the optimal revascularization method for patients with left main coronary artery disease. Generally, the higher the SYNTAX score, the greater the likelihood that the patient will be qualified for CABG, although the qualification score itself generally depends on many factors related to chronic comorbidities and other angiographic parameters, not just the SYNTAX score. We added in Discussion section: “Generally, the higher the SYNTAX score, the greater the likelihood that the patient will be qualified for CABG, although the qualification score itself depends on many factors related to chronic comorbidities and other angiographic parameters, not just the SYNTAX score.” Given that patients in the PCI cohort had a significantly higher co-disease burden than those in the CABG group, many patients with high SYNTAX scores were considered candidates for PCI due to their high surgical risk. Therefore, in our study, SYNTAX scores were comparable between both cohorts.

Your inquiry: Although frailty was defined using the Cardiovascular Health Study criteria, the manuscript does not detail the scoring method or how these criteria were applied operationally. Authors should clarify this aspect

Our response: Frailty score was assessed using Rockwood Frailty Scale and we assessed each patient personally using this scale. We added in Materials and Methods section: “and assessed using Rockwood Frailty Scale (each patient was assessed personally).”

Your inquiry: The definition of “repeat revascularization” requires clarification. It is not specified whether this includes only unplanned procedures or also planned staged PCI interventions.

Our response: The “repeat revascularization (RR)” refers to unplanned procedures and also planned staged PCI interventions – both target-vessel or target-lesion driven, both clinically and angiographic driven. We added in Materials and Methods section: “The RR refers to unplanned procedures and also planned staged PCI interventions - both target-vessel or target-lesion driven, both clinically and angiographic driven.”

Your inquiry: Further details should be provided on how the Heart Team decisions were standardized, including whether a checklist, algorithm, or structured scoring system was employed.

Our response: We added in Materials and Methods section: “All patients presented during the Heart Team meetings were provided with a personalized patient history file containing their medical history, coronary angiography, echocardiography, and laboratory test results. Each Heart Team member had access to this file and the opportunity to review it. Subsequently, after the presentation of the angiography and echocardiography results, the Heart Team members discussed and made a decision regarding eligibility. Only a unanimous decision by all Heart Team members constituted final approval for further treatment.”

Your inquiry: The logistic regression model used for propensity score matching should list the specific covariates included, particularly whether the SYNTAX score was among them, which is not clear

Our response: The covariates used for propensity score matching were included in Table S1 added in the end of TABLES section. SYNTAX score are among them.

Table S1. Baseline clinical and angiographic characteristics in the propensity-score–matched cohorts.

Propensity-score–matched patients

CABG (53)

PCI (53)

CLINICAL CHARACTERISTICS

Age, years; median (Q1;Q3)

70.8 (61.8;75.7)

71.0 (61.0;76.0)

Gender, male; n(%)

41 (77.4)

42 (79.2)

BMI, kg/m2; median (Q1;Q3)

28.2 (26.9; 29.1)

28.1 (26.7;28.8)

Frailty; n(%)

4 (7.5)

4 (7.5)

Current smoking; n(%)

13 (24.5)

12 (22.6)

COPD; n(%)

9 (17.0)

9 (17.0)

Diabetes; n(%)

20 (37.7)

19 (35.8)

       with insulin; n(%)

9 (17.0)

8 (15.1)

Hypertension; n(%)

44 (83.0)

46 (86.8)

Severe PH; n(%)

7 (13.2)

7 (13.2)

Dyslipidemia; n(%)

39 (73.6)

39 (73.6)

Congestive heart failure; n(%)

10 (18.9)

11 (20.8)

       LVEF, %; median (Q1;Q3)

34.1 (31.2;50.1)

33.7 (30.8; 49.6)

CKD; n(%)

15 (28.3)

16 (30.2)

Atrial fibrillation; n(%)

13 (24.5)

13 (24.5)

Anemia; n(%)

15 (28.3)

17 (32.1)

Prior MI; n(%)

27 (50.9)

28 (52.8)

Prior revascularization; n(%)

19 (35.8)

19 (35.8)

Indication; n(%)

    CCS

22 (41.5)

20 (37.7)

    ACS

31 (58.5)

33 (62.3)

       STEMI

1 (1.9)

2 (3.8)

       NSTEMI/UA

30 (56.6)

31 (58.5)

PAD; n(%)

5 (9.4)

5 (9.4)

Prior stroke/TIA; n (%)

7 (13.2)

8 (15.1)

Active cancer; n(%)

0 (0.0)

3 (5.7)

EuroSCORE II, %; median (Q1;Q3)

3.8 (3.3;5.2)

3.9 (3.1;5.5)

STS score, %; median (Q1;Q3)

2.5 (2.2;3.2)

2.6 (2.0;3.4)

ANGIOGRAPHIC PARAMETERS

Qualifying LM lesion

   LM coronary segment

52 (98.1)

51 (96.2)

   LM equivalent disease

1 (1.9)

2 (3.8)

Bifurcation or trifurcation disease of distal LM; n(%)

46 (86.8)

44 (83.0)

LM considered as culprit lesion in ACS, n(%)

5/31 (16.1)

6/33 (18.2)

Affected non-LM arteries, n(%)

0

9 (17.0)

10 (18.9)

1

13 (24.5)

14 (26.4)

2

18 (34.0)

17 (32.1)

3

13 (24.5)

12 (22.6)

Severe calcification; n(%)

19 (35.8)

18 (34.0)

Total occlusion; n(%)

13 (24.5)

13 (24.5)

SYNTAX score; median (Q1;Q3)

34.1 (29.2;41.4)

34.0 (28.9;41.2)

Your inquiry: The reported range for hospital stay appears inconsistent; for example, a median duration of 6 days with an interquartile range (IQR) of 8–11 days is statistically implausible, as the median should fall between the first and third quartiles. Authors should clarify this aspect.

Our response: This was incorrect entry. Postprocedural hospital stay, days; median (Q1;Q3) in CABG-cohort was corrected to 6 (4;11) both in Results Outcomes section as well as in Table 4.

Your inquiry: Moreover, authors are encouraged to include the latest evidence about the different prognosis of ACS patients (https://doi.org/10.1007/s40256-025-00739-8)

Our response: We added in Discussion section: “Additionally, in a recently published article regarding one-year prognostic differences and management strategies between ST-elevation and non-ST-elevation myocardial infarction at 1-year follow-up, overall outcomes were comparable between groups. Nonfatal reinfarction occurred more frequently in patients with NSTEMI (3.4% versus 2.8%, P=0.022), but this association was not significant after adjustment (odds ratio [OR] 0.90, 95% confidence interval [CI] 0.65–1.24, P=0.519). Results from propensity score-matched analyses confirmed the absence of prognostic differences. Subgroup analyses revealed significant interactions for diabetes mellitus and completeness of revascularization. As the authors of the article emphasized 1-year outcomes were largely similar in patients with STEMI and NSTEMI. Differences in reinfarction risk appear to be driven by baseline characteristics and treatment patterns, rather than in-farct type itself. [32]” We also added new reference in References section regarding this article - 32. Spadafora L, Pastena P, Cacciatore S, et al. One-Year Prognostic Differences and Management Strategies between ST-Elevation and Non-ST-Elevation Myocardial Infarction: Insights from the PRAISE Registry. Am J Cardiovasc Drugs. 2025; Jun 24. Online ahead of print.

That is why reference previously no. 32. “Abdulrahman M, Alsabbagh A, Kuntze T, et al. Impact of Hierarchy on Multidisciplinary Heart-Team Recommendations in Patients with Isolated Multivessel Coronary Artery Disease. J Clin Med. 2019; 8.” is now reference no. 33. We corrected the number in Limitations and References section.

Reviewer 2 Report

Comments and Suggestions for Authors

General comment:
This is a clinically meaningful and well-structured study addressing a persistent question in interventional cardiology: how to best approach revascularization in patients with left main coronary artery (LMCA) disease and multivessel disease (MVD) in routine clinical practice. The authors provide real-world data from a high-volume center, with decisions made through a Heart Team (HT) process, which adds value to the field.

Strengths

  1. Real-life data in a well-defined population:
    The study’s focus on an all-comer cohort assessed by a multidisciplinary HT fills a notable gap in the literature, particularly as many RCTs exclude frail or complex patients. This enhances external validity.

  2. Robust statistical analysis:
    Use of propensity score matching strengthens the comparisons and helps mitigate confounding. The presentation of both unmatched and matched results, including hard endpoints, is commendable.

  3. Mature long-term follow-up:
    A median of 5.5 years follow-up allows for a meaningful assessment of outcomes such as mortality, MI, stroke, and repeat revascularization.

  4. Clarity and structure:
    The manuscript is well written, with logical flow and transparent reporting. The figures and tables are informative and adequately support the text.

  5. Balanced discussion:
    The authors avoid overstatement and acknowledge the limitations of their observational design. The discussion is comprehensive, well referenced, and places findings in context with key trials (EXCEL, NOBLE, SYNTAX, PRECOMBAT, etc.).

Points for Improvement

  1. MACCE definition and RR rates:
    While the higher MACCE and RR rates in the PCI group are expected, it would be helpful to explicitly state whether all RR events were target-vessel or target-lesion driven. Were they clinically driven, or angiographic? This could impact interpretation significantly.

  2. Frailty assessment:
    Frailty is increasingly recognized as a key factor in revascularization decisions. The authors mention using CHS criteria, but further details on how frailty was operationalized and whether it influenced HT decision-making (beyond statistical correlation) would enrich the manuscript.

  3. Completeness of revascularization:
    The CABG group achieved higher complete revascularization rates. The impact of this on MACCE and MI should be further emphasized, potentially via a subgroup analysis or discussion of residual SYNTAX score.

  4. Antiplatelet therapy duration and strategy:
    Since PCI patients were more likely to receive P2Y12 inhibitors, some mention of DAPT duration and any heterogeneity in therapy (clopidogrel vs. ticagrelor/prasugrel) would be informative. This may have influenced bleeding or ischemic events, even if not captured directly.

  5. Role of intracoronary imaging:
    IVUS use was excellent (>97%), which likely contributed to procedural success in PCI patients. A brief note in the discussion about how this may differ from centers with lower imaging uptake could help readers interpret generalizability.

  6. Presentation of stroke risk:
    The stroke difference (CABG > PCI) is statistically significant and important. It might be useful to explore whether any perioperative factors (e.g., aortic manipulation, clamping strategy) were captured or whether stroke timing was perioperative vs. long-term.

Minor Comments

  • Abstract: It is dense but informative. Consider clarifying that the HT process was prospective, even if the analysis is retrospective.

  • Table 4: You may consider bolding statistically significant p-values for clarity.

  • Grammar: Minor edits would improve fluency (e.g., “what led to never-ending debate” could be rephrased as “which has led to ongoing debate”).

Conclusion and Recommendation

This is a well-executed, clinically relevant observational study providing valuable insight into real-world LMCA revascularization outcomes. It reinforces the importance of HT-based decision-making and highlights key differences in secondary outcomes, especially MACCE and stroke.

Recommendation: MAJOR revision

Author Response

Dear Reviewer

We would like to thank the Editors and the Reviewers for taking the time to evaluate this work and for providing valuable comments, which, as we believe, helped us to further improve the submitted manuscript.

The introduced changes were highlighted in the manuscript in yellow.

Your inquiry: MACCE definition and RR rates: While the higher MACCE and RR rates in the PCI group are expected, it would be helpful to explicitly state whether all RR events were target-vessel or target-lesion driven. Were they clinically driven, or angiographic? This could impact interpretation significantly.

Our response: The “repeat revascularization (RR)” refers to unplanned procedures and also planned staged PCI interventions – both target-vessel or target-lesion driven, both clinically and angiographic driven. We added in Materials and Methods section: “The RR refers to unplanned procedures and also planned staged PCI interventions - both target-vessel or target-lesion driven, both clinically and angiographic driven.”

Your inquiry: Frailty assessment: Frailty is increasingly recognized as a key factor in revascularization decisions. The authors mention using CHS criteria, but further details on how frailty was operationalized and whether it influenced HT decision-making (beyond statistical correlation) would enrich the manuscript.

Our response: Frailty score was assessed using Rockwood Frailty Scale and we assessed each patient personally using this scale. We added in Materials and Methods section: “and assessed using Rockwood Frailty Scale (each patient was assessed personally).” Generally, when patient was assessed as frail, Heart Team was compliant for qualification to PCI.

Your inquiry: The CABG group achieved higher complete revascularization rates. The impact of this on MACCE and MI should be further emphasized, potentially via a subgroup analysis or discussion of residual SYNTAX score.

Our response: We added in Discussion section: “Also, the higher rates of complete revascularization in CABG cohort was likely associated with higher MI- and RR-rates in long-term follow-up in PCI group.”

Your inquiry: Since PCI patients were more likely to receive P2Y12 inhibitors, some mention of DAPT duration and any heterogeneity in therapy (clopidogrel vs. ticagrelor/prasugrel) would be informative. This may have influenced bleeding or ischemic events, even if not captured directly.

Our response: This is very significant comment. We added data about usage of clopidogrel, ticagrelor or prasugrel in Table 3. Generally, the DAPT duration was consistent with ESC guidelines for chronic and acute coronary syndromes – for ACS – optimally 12 months, for CCS – optimally 6 months.

P2Y12 inhibitors; n(%)

145 (54.3)

21 (19.3)

124 (78.5)

<0.001

       Clopidogrel, n(%)

88 (60.7)

19 (90.5)

69 (55.6)

<0.001

       Ticagrelol, n(%)

37 (25.5)

1 (4.8)

36 (29.0)

<0.001

       Prasugrel, n(%)

20 (13.8)

1 (4.8)

19 (15.3)

<0.001

Your inquiry: IVUS use was excellent (>97%), which likely contributed to procedural success in PCI patients. A brief note in the discussion about how this may differ from centers with lower imaging uptake could help readers interpret generalizability.

Our response: We added in Discussion section: “The high utilization of intracoronary imaging modalities in our study may have translated into a high rate of successful percutaneous interventions. PCI outcomes in other centers with lower rates of intracoronary imaging usage would likely have been worse.”

Your inquiry: The stroke difference (CABG > PCI) is statistically significant and important. It might be useful to explore whether any perioperative factors (e.g., aortic manipulation, clamping strategy) were captured or whether stroke timing was perioperative vs. long-term.

Our response: We have added in Table 4 data about rates of periprocedural strokes:

      Periprocedural, n(%)

4 (1.5)

3 (2.8)

1 (0.6)

0.27 (0.10-0.75)

0.01

We have no data about perioperative factors (e.g., aortic manipulation, clamping strategy).

Your inquiry: Minor Comments

Abstract: It is dense but informative. Consider clarifying that the HT process was prospective, even if the analysis is retrospective.

Our response: We added in Methods and Material Section “Heart Team meetings were prospective, while the analysis was retrospective.”

Your inquiry: Table 4: You may consider bolding statistically significant p-values for clarity.

Our response: It was corrected.

Your inquiry: Grammar: Minor edits would improve fluency (e.g., “what led to never-ending debate” could be rephrased as “which has led to ongoing debate”).

Our response: It was corrected.

Round 2

Reviewer 1 Report

Comments and Suggestions for Authors

Congratulations to the authors for having implemented all of my comments into their manuscript.

Author Response

Dear Reviewer

Thank you for reviewing this manuscript.

Reviewer 2 Report

Comments and Suggestions for Authors

Suggestions

  1. Language and grammar: Although significantly improved, some passages would benefit from additional language polishing. Occasional awkward phrasing (e.g., “This shift might be due to significant technical advancements…”) could be further refined for clarity and flow.

  2. Terminology consistency: Occasionally, the manuscript switches between “percutaneous stenting” and “PCI”. Consider harmonizing terminology throughout.

  3. Figures and tables: Ensure that all figure captions are self-explanatory and include appropriate abbreviations. For instance, in Figure 3, a brief explanation of subgroups would aid reader understanding.

  4. ESC Guidelines integration: While references to ESC guidelines are appropriate, the authors may briefly elaborate on how their findings support or challenge current ESC 2018 recommendations in the conclusion.

  5. Limitations section: This section is candid and appropriate, but the point on Heart Team hierarchy bias could benefit from more elaboration or referencing.

Comments on the Quality of English Language

minor polishing

Author Response

Dear Reviewer

Thank you for reviewing this manuscript. We provide version with improved English style as much as we could.
